# Current Progress in Lipidomics of Marine Invertebrates

**DOI:** 10.3390/md19120660

**Published:** 2021-11-25

**Authors:** Andrey B. Imbs, Ekaterina V. Ermolenko, Valeria P. Grigorchuk, Tatiana V. Sikorskaya, Peter V. Velansky

**Affiliations:** 1Laboratory of Comparative Biochemistry, Zhirmunsky National Scientific Center of Marine Biology, Far Eastern Branch, Russian Academy of Sciences, 17 Palchevskogo Str., 690041 Vladivostok, Russia; andrey_imbs@hotmail.com (A.B.I.); ecrire_711@mail.ru (E.V.E.); Miss.tatyanna@yandex.ru (T.V.S.); 2Laboratory of Cell Biology, Federal Scientific Center of the East Asia Terrestrial Biodiversity, Far Eastern Branch, Russian Academy of Sciences, 159 Pr-t 100-let Vladivostoka Str., 690022 Vladivostok, Russia; kera1313@mail.ru

**Keywords:** mollusks, cnidarians, echinoderms, crustaceans, holothurians, fatty acids, phospholipids, lipid molecular species, mass spectrometry, lipidomics

## Abstract

Marine invertebrates are a paraphyletic group that comprises more than 90% of all marine animal species. Lipids form the structural basis of cell membranes, are utilized as an energy reserve by all marine invertebrates, and are, therefore, considered important indicators of their ecology and biochemistry. The nutritional value of commercial invertebrates directly depends on their lipid composition. The lipid classes and fatty acids of marine invertebrates have been studied in detail, but data on their lipidomes (the profiles of all lipid molecules) remain very limited. To date, lipidomes or their parts are known only for a few species of mollusks, coral polyps, ascidians, jellyfish, sea anemones, sponges, sea stars, sea urchins, sea cucumbers, crabs, copepods, shrimp, and squid. This paper reviews various features of the lipid molecular species of these animals. The results of the application of the lipidomic approach in ecology, embryology, physiology, lipid biosynthesis, and in studies on the nutritional value of marine invertebrates are also discussed. The possible applications of lipidomics in the study of marine invertebrates are considered.

## 1. Introduction

Invertebrates have not been recognized as an actual taxon since this group includes all animals that do not have a spine and that were placed in it by the so-called residual principle. This is a classic example of a paraphyletic group. According to modern concepts, the animals attributed by Lamarck to invertebrates are divided into more than twenty equivalent groups of a higher rank, phyla. Furthermore, there are numerous departments of invertebrate zoology, and the term “invertebrates” currently indicates the professional specialization of zoologists. The group of marine invertebrates comprises more than 90% of all marine animal species [1].

Lipids are among the most important organic compounds found in all marine invertebrates. Lipids form the structural basis of cell membranes, are utilized as an energy reserve by these animals, and are, therefore, used as important indicators of their ecology and biochemistry. Lipids are relatively small hydrophobic molecules and are highly diverse in their structure. There are several definitions of “lipids”, but we focus mainly on fatty acids (FAs) and lipids with fatty acyl groups such as wax esters (WEs), triacylglycerols (TGs), monoalkyldiacylglycerols or diacylglycerol ethers (DAGEs), glycerophospholipids (GPLs), and sphingolipids.

Due to the rapid development of thin-layer chromatography, gas chromatography on capillary columns coupled with mass spectrometric detectors, high-resolution nuclear magnetic resonance, and preparative high-performance liquid chromatography, detailed information on the lipid class composition and FA composition of total lipids/lipid classes of marine invertebrates has become increasingly available, not only for the order/family level but also for the genus/species level. However, this classic lipidology deals with integral FA and lipid data because each lipid class consists of hundreds of lipid molecules, and total FAs are obtained through hydrolysis of thousands of lipid molecules.

Lipid molecules of certain types are often referred to as “lipid molecular species”; the term “lipidome” describes the complete lipid profile of an organism, and the studies in lipidomics consider the structure, function, interaction, and dynamics of lipid molecular species [2]. The field of lipidomics was first defined in 2003 [3] through integrating the defined chemical properties in individual lipid molecules with a comprehensive approach based on mass spectrometry.

The development of high-performance liquid chromatography and mass spectrometry has made it possible to analyze molecular species of lipids without any sample preparation. There are two main approaches to lipidome analysis: untargeted and targeted. Untargeted lipidomics implies the determination of the complete lipid profile using high-resolution mass spectrometers, including those without chromatographic separation (shotgun lipidomics). Targeted lipidomics focuses on the precise quantification of specific lipid molecules. In this case, sensitive high-speed mass spectrometers (triple quadrupole or quadrupole linear-ion trap) are usually used in the multiple-reaction monitoring mode.

During the period covered by this review, new equipment and methods have appeared that allow lipidomic studies to be carried out at a higher level. In 2005, mass spectrometers with a new type of high-resolution analyzer (Orbitrap) have become commercially available. Since 2006, ion-mobility mass spectrometers have become available, designed for better analysis of small molecules through separation based on the conformation of a molecule in addition to its mass. New methods that allow the determination of double-bond positions in specific FAs at a specific sn-position using MS with ozone-induced dissociation (OzID or O_3_-MS) [4,5] have appeared.

Compared to classical lipidology, the lipidomic approach provides more information about the lipid composition (distribution and sn-position in the glycerol backbone of acyl, alkyl, and alkenyl groups) and allows for more accurate quantitative analysis. Thus, the researcher receives a much larger amount of data that can be used to solve a wide range of problems in the study of invertebrates: lipid biosynthesis pathways determination, chemotaxonomy, determination of biotic and abiotic factors effects, investigation of embryogenesis, ontogenesis, and food chains.

In contrast to FAs and lipid classes, lipid molecular species profiles of marine invertebrates still remain very poorly understood. However, the number of publications on lipidomics of these marine animals showed an explosive increase over the past five years (Figure 1). This review highlights the progress in lipidomics of marine invertebrates for the period of 2004–2021.

## 2. Research Objects

The largest phyla of marine invertebrates in terms of the number of species are mollusks, cnidarians, arthropods, echinoderms, worms, and sponges. The group of marine invertebrates comprises more than a million species, but lipidomes are at least partially known for less than one hundred of them (Table 1). Total lipidomes including both structural and storage lipids have been described from only a few species.

The phylum Mollusca comprises more than 110,000 species. The best known mollusks are oysters, abalones, clams, scallops, sea slugs, octopus, and squid. It is not surprising that the first lipidomic investigations of mollusks were conducted on commercial edible species, which are wild-harvested and cultivated in many countries. Initially, the lipidomes of five oyster species were analyzed: *Crassostrea hongkongensis* [6], *C. lugubris* [8], *C. plicatula* [7], *C. talienwhanensis* [10], *Ostrea gigas* [11], and *O. edulis* [12]. The composition of lipid molecular species was determined in mussels, *Mytilus galloprovincialis* [13,14,15,16], and their edible feet [12], and also in adult [11,17,18,19] and juvenile blue mussels, *M. edulis* [20]. The lipidome was examined in 11 clam species such as *Corbicula fluminea* [21,22], *Cyclina sinensis* [23], *Mactra chinensis* [23], *M. veneriformis* [23], *Meretrix lyrate* [24], *M. meretrix* [23], *Ruditapes philippinarum* [17,23,25], *Saxidomus purpurata* [23], *Sinonovacula constricta* [11,26], *Solen gouldi* [26], *Venerupis variegate* [11], and in the edible foot of *Venus gallina* [12]. Two abalone species (*Haliotis rubra* [11] and *H. discus hannai* Ino [17,27]) were also investigated. Lipids of three scallop species (*Chlamys islandica* [28], *Patinopecten yessoensis* [17], and *Placopecta magellanicus* [11]) were studied by the lipidomic approach. Molecular species of GPL and phosphonolipids were analyzed in the deep-sea nudibranch mollusks *Tritonia tetraquetra*, *Dendronotus* sp., and *D. robustus* [29,30]. The lipidomes were described from the sacoglossan sea slugs *Elysia viridis* and *Placida dendritica,* which are able to sequester chloroplasts from algae and incorporate them into their cells [31]. The lipid molecular species of TG in the squid *Berryteuthis magister* [32] and polar lipids of squid viscera and gonads were characterized [33,34].

The phylum Cnidaria comprises about 30,000 species. The best-studied cnidarians are corals, sea fans, and sea anemones belonging to the class Anthozoa, and also hydrocorals (Hydrozoa) and jellyfish (Scyphozoa). Most lipidomic studies of cnidarians were based on the coral species that form the structural basis of coral reefs. Total lipidomes were analyzed in the reef-building coral *A. cerealis* [36], three alcyonarians (*Sinularia* sp. [43], *S. heterospiculata* [44], and *S. siaesensis* [45]), the zoantharian *Palythoa* sp. [50], and three ascidians (*Ciona intestinalis*, *Halocynthia roretzi*, and *Styela clava* [51]). Compositions of WE molecular species were studied in reef-building corals (*Acropora acuminata*, *A. cytherea*, and *Turbinaria peltata*), asymbiotic tropical soft corals (*Mopsella spinosa*, *Menella flora*, and *M. praelonga*), and symbiotic tropical soft corals (*Cladiella laciniosa*, *Sinularia brassica*, and *S. robusta*) [35]. The TG and DAGE composition of the hydrocoral *M. platyphylla* has recently been published [49]. Some lipid molecular species were identified in the metabolomes of *A. cervicornis* [37] and *Pocillopora damicornis* [38]. Polar lipidomes were characterized in the reef-building coral *Seriatopora caliendrum* [39,40], the asymbiotic cold-water soft coral *Gersemia rubiformis* [38], and several species of symbiotic tropical soft corals such as *Capnella* sp. [42], *Xenia* sp. [46], and *Sinularia macropodia* [42]. The polar lipid molecular species were analyzed in two symbiotic tropical hydrocorals (*Millepora dichotoma* and *M. platyphylla*) and the asymbiotic cold-water hydrocoral *Allopora steinegeri* [47,48,49]. Two jellyfish species (*Rhopilema esculentum* [52] and *Phyllorhiza punctata* [53]) and the sea anemone *Aiptasia pallida* [54,55] were also subjected to lipidomic analyses.

Most tropical species of corals, hydrocorals, zoantharians, sea anemones, and jellyfish contain intracellular symbiotic dinoflagellates of the family Symbiodiniaceae (referred to as zooxanthellae). These microalgae are essential for the life of host animals, and their lipid molecular species were described as part of the total lipidome of symbiotic cnidarians [43,54,55,75,76,77,78,79].

The Decapoda is an order of crustaceans within the class Malacostraca (the phylum Arthropoda). This order comprises nearly 15,000 species, including such well-studied groups as crabs, shrimp, prawns, and lobsters. Many of these crustaceans are target species of commercial fisheries and mariculture. Therefore, the lipidomic studies on crustaceans have mainly dealt with crab and shrimp species. The profile of lipid molecular species was analyzed in the juvenile Dungeness crab *Metacarcinus magister* (= *Cancer magister*) [56], embryos of the crabs *Carcinus maenas* and *Necora puber* [57,58], the hepatopancreas of the Chinese mitten crab *Eriocheir sinensis* [59,60] and the swimming crab (also known as the *gazami* crab, South Korea’s blue crab, or horse crab) *Portunus trituberculatus* [62], and also edible viscera and muscles of the crabs *P. trituberculatus* and *P. pelagicus* [61]. Among shrimp species, *Macrobrachium borellii* [64], juvenile whiteleg shrimp *Litopenaeus vannamei* [63]; deep-sea hydrothermal vent shrimp *Mirocaris fortunata* [65]; the muscle and cephalothorax of *Penaeus kerathurus* [66,67]; and the head and body carapace of *P. monodon*, *P. vannamei*, and *P. chinensis* [68] were studied. Lipidomes were also analyzed in the muscles and cephalothorax of the spiny lobster *Palinurus vulgaris* [69], copepods *Calanus* sp. [70], and in the waste produce of seafood processing (shrimp heads) [34].

The phylum Echinodermata comprises about 6,000 species. The best-known echinoderms are sea cucumbers (the class Holothuroidea), starfishes (the class Asteroidea), sea urchins (the class Echinoidea), and brittle stars (the class Ophiuroidea). The number of sea cucumber (or holothurian) species amounts to about 1700. Many of these are edible and some are culturable. Lipidome analysis was performed on 12 holothurian species: *Apostichopus japonicus* [71], *Actinopyga mauritiana* [72], *Bohadschia marmorata* [72,73], *Cucumaria frondosa* [71,73], *Eupentacta fraudatrix* [71], *Isostichopus fuscus* [73], *Holothuria atra* [72], *H. edulis* [72], *H. leucospilota* [72], *H. mexicana* [73], *H. polii* [72,73], and *Parastichopus califormicus* [73]. About 1900 starfish species inhabit the ocean seabed, but the composition of the major phospholipid molecular species is currently known for only two species, *Distolasterias nipon* and *Asterias amurensis* [71]. To date, the polar lipidome of one sea urchin species, *Strongylocentrotus intermedius*, has been described [71].

Between 9000 and 15,000 species of sponges have been classified within the phylum Porifera. Sponges are not edible to humans, and, to date, only one species, *Amphimedon queenslandica*, has been studied by the lipidomic approach [74].

Thus, a comparison between the number of species of marine invertebrates with at least partially identified lipidomes and the number of all invertebrate species has shown that lipidomics of marine invertebrates will provide plenty of research materials in the coming decades.

## 3. Lipidomic Data

Since the early 1950s, the contents of total lipids and lipid classes, as well as the FA composition of total, non-polar (storage), and polar (structural) lipids of marine invertebrates have been examined by a large number of laboratories across the world. In some cases, FA composition of certain lipid classes has been analyzed. As a result, a general concept of distribution of lipid classes and FA over these animals, from genera to phyla, has been formed. At the beginning of lipidomic studies, several intermediate investigations analyzed the FA composition of each lipid class but not their actual molecular species profiles [53,66,67]. Currently, many studies deal with the polar part of lipidomes (mainly GPL); however, some works have already presented total lipidomes [43,44,45,50]. In this part, we have attempted to characterize the major lipid molecular species and their chemotaxonomic distribution in the marine invertebrates studied (Table 1).

In marine invertebrates, the major non-polar lipid classes are TG, WE, and DAGE. The major polar lipid classes are phospholipids: PC, PE, PS, and PI. Accordingly, these four GPL classes contain choline, ethanolamine, serine, and inositol groups in the polar head of their molecules. Depending on the groups at *sn*-1 and *sn*-2 positions of the glycerol backbone, each GPL class can contain three forms (or subclasses): 1,2-diacyl, 1-*O*-alkyl-2-acyl (plasmanyl), and 1-*O*-(alk-1′-enyl)-2-acyl (plasmenyl, PlsGPL). After the name of ether moiety (number of carbon atoms: number of double bonds) at *sn*-1 position, the letter “a” indicates 1-*O*-alkyl group, and the letter “p” indicates 1-*O*-(alk-1′-enyl) group. In addition to GPL, ceramide aminoethylphosphonate (CAEP), a sphingosine-based phosphonolipid, is one of the major structural lipids in marine invertebrates.

### 3.1. Mollusca

In mollusks, PC is the most abundant GPL class, followed by PE. Up to 290 molecular species of polar lipids (PL) and up to 1200 molecular species of total lipids (TL) can be found in oysters *Crassostrea* and *Ostrea* [6,7,8,9,10,11,12]. The most abundant molecular species of oyster GPL have the diacyl form and contain saturated FAs (mainly 16:0 and 18:0) at the *sn*-1 position and polyunsaturated FAs (mainly, 20:5n-3 and 22:6n-3) at the *sn*-2 position of the glycerol backbone. For example, PC 16:0/20:5, PC 16:0/22:6, PE 16:0/20:5, PE 18:0/20:5, PS 18:0/20:5, and PI 18:0/22:6 are the most abundant species of diacyl GPL in *C. plicatula* [7]. Among alkylacyl-GPL, PC 16:0a/22:6, PE 18:0a/20:5, and PS 18:0a/22:2 are the dominant molecular species. The molecules with 18:0p/22:6 and 18:0p/22:2 may dominate the alkenylacyl-GPL, PE, and PS, respectively. The molecules with non-methylene-interrupted (NMI) FAs such as PS 18:0p/22:2, PS 20:0p/20:2, and PS 16:0p/22:2 are also observed in *C. gigas* [11]. The non-polar part of the oyster lipidome has not been analyzed.

About 410 molecules in TL [19], 230 molecules of PL [18], and 40 molecules of TG [20] have been identified in mussels of the genus *Mytilus*. All major ethanolamine GPL recognized in mussel lipids are plasmalogens with the 18:0 chain dominating these ether PL [12,13]. Besides 20:5n-3 and 22:6n-3, an abundance of NMI acyl chains has been found in mussel alkenylacyl-PE [15,18]. Similar to oyster lipids, the PC of mussels is mainly composed of diacylic molecular species, with the 20:5n-3 and 22:6n-3 chains prevailing at the *sn*-2 position of glycerol [15]. However, Wang et al. [11] detected 33, 54, and 63% of the alkenylacyl form in the PC, PE, and PS of *M. edulis*, respectively. A high abundance of NMI fatty acids, like that of the 20:2 and 22:2 ones, is found in alkenylacyl-PS [11]. Polar lipidomes of adult and juvenile *M. edulis* are similar; cardiolipin (CL) and their tetradocosahexaenoic molecular species (CL 88:24) have additionally been recorded from juveniles [20]. Acyl chains of TG molecules in juveniles of *M. edulis* contain up to 62 carbon atoms and up to 15 double bonds [20].

The lipidome of the blue mussel *M. galloprovincialis* has been intensively investigated. In a general analysis, 14 WE, 34 TG, 41 PC, 23 PE, 4 PS, 6 PI, and 10 CAEP molecular species were detected in the total lipids of this animal [13]. However, direct analyses of PL allowed a detailed description of 185 choline GPL, 131 ethanolamine GPL, 45 lysoPE, 57 lysoPC, 33 CAEP, 14 *N*-monomethylated CAEP, and 19 CPE (ceramide phosphoethanolamines) molecules [14,15,16]. Mussels of this species are well known to contain at least four classes of ceramide lipids (CAEP, *N*-Me-CAEP, *N*-diMe-CAEP, and CPE), but only a few studies describe the variety of bivalve ceramide molecular species [14]. Trace amounts of unusual diacyl choline and ethanolamine phosphonolipids have been found, which lack the oxygen atom between the phosphorus atom and choline or ethanolamine groups [13,15]. Another interesting acyl chain found at the *sn*-2 position of alkenylacyl-PE is 28:8 [15].

To date, about 450 molecular species of GPL have been recorded from clam lipidomes [23,26]. Considerable differences are found in the composition of predominant molecular species between the diacyl, alkylacyl, and alkenylacyl forms of each GPL class [23]. Similarly to the major GPL molecules in oysters and mussels, those in clams are characterized by the dominance of 20:5 and 22:6 acyl groups at the *sn*-2 position of the glycerol backbone. Noticeable amounts of a less unsaturated PE (16:0/18:2 and 18:0/18:2) and PC 16:0/18:1 have been recorded from the clam *V. gallina* [12], and PE 18:0p/18:1 from the clam *M. lyrata* [24]. Clam PL is reported to contain minor amounts of phosphatidylglycerol (PG) [26]. Several studies show that the proportion of alkenylacyl forms ranges from 23 to 64% in three phospholipid classes (PE, PC and PS) of these clams [11,21,22,23]. However, Boselli et al. [12] described only alkenylacyl-PE molecular species, while Tran et al. [24] found only alkenylacyl-PE and alkenylacyl-PS molecular species in clams. The TG profile of the clam *R. philippinarum* has been published [25].

A total of about 125 lipid molecular species have been identified in the lipidome of the abalone *H. discus hannai* Ino [17], of which TG 16:0/18:0/20:4, TG 16:0/18:2/18:2, PE 18:4/20:5, PE 18:2/22:6, and phosphatidic acid, PA 18:0/18:2, are the most abundant molecular species in viscera and gonads [27]. A considerable level of alkenylacyl forms of PE, PC, and PS is found in PL of the abalone *H. rubra* [11].

The lipidomic approach has allowed the identification of 224 lipids in the digestive gland of the scallop *C. islandica*, including diacyl, alkylacyl, and alkenylacyl-PC; diacylglycerols (DG) and TG have also been analyzed [28]. PC 36:5 PC 38:6, alkylacyl-PC 38:6/alkenylacyl-PC 38:5, and alkenylacyl-PC 38:6 mainly contribute to the molecular species of choline GPL of *Chlamys’* digestive gland [28]. PC 16:0/20:5, PC 16:0/22:6, PE 16:0/20:5, PE 18:0/20:5, PE 16:0/22:6, and PI 18:0/20:5 are the major PL molecules in the scallop *P. yessoensis* [17]. Both PL profiles in the scallop *P. magellanicus* were studied by Wang et al. [11]. Gilaber et al. [28] showed the detail profiles of TG and DG in the digestive gland of C. *islandica*, where 102 TG and 51 DG molecular species were identified. TGs of 52 to 58 carbon atoms in their acyl chains and of 5 to 11 double bonds, together with DGs of 36 to 44 carbon atoms and 5 to 11 double bonds, are the dominant neutral glycerolipids.

The cold-water nudibranch *D. robustus* feeds mainly on detritus, and common phospholipid molecules of this mollusk contain the 20:5 and 20:4 acyl groups at the *sn*-2 position [30]. In contrast to *D. robustus*, the cold-water nudibranchs *T. tetraquetra* and *Dendronotus* sp. prey on soft corals, which are a source of very-long-chain FA (VLCFA), and, therefore, contain a considerable part of PE, PC, and PS molecular species with C_24_ polyunsaturated FA (PUFA) including, e.g., PE (16:0p/24:5 and 16:0p/24:6), PC (16:0a/24:5 and 16:0a/24:6), and PS (16:0a/24:5 and 18:0a/24:5) [29,30]. A large portion of isobaric molecular species (with the same molecular weight) can be found in phospholipids of both the nudibranchs feeding on soft corals and the nudibranch feeding on detritus. The former contain C_16_ alkyl/C_24_ acyl molecules, while the latter contain C_20_ alkyl/C_20_ acyl or C_18_ alkyl/C_22_ acyl molecules [30]. All molecular species of CAEP of the nudibranchs contain only 16:0 as the *N*-acyl group.

In addition to the common GPL of mollusks, the molecular species of monogalactosyldiacylglycerol (MGDG), monogalactosylmonoacylglycerol (MGMG), digalactosyldiacylglycerol (DGDG), digalactosylmonoacylglycerol (DGMG), diacylglyceryltrimethylhomoserine (DGTS), monoacylglyceryltrimethylhomoserine (MGTS), sulfoquinovosyldiacylglycerol (SQDG), and sulfoquinovosylmonoacylglycerol (SQMG) are found in the lipidome of the sacoglossan sea slugs *E. viridis* and *P. dendritica* [31]. These lipid classes are characteristic of the plastidial membranes and are transferred to mollusks when algae chloroplasts are sequestered and incorporated into mollusk’s cells.

A total of 1,223 molecular species of PL have been identified and quantified in squid viscera and gonads (byproducts of seafood processing) [33]. In squid gonads, the major PL classes are PC and PE, including PC 16:0/22:6, PC 16:0/18:1, PC 16:0/20:5, PC 16:0/20:1, PC 18:0/22:6, PE 20:1/20:5, PE 16:0/20:5, PE 18:0/20:5, and PE 20:1/22:6 [34]. In contrast to the neutral lipids of bivalve and gastropod mollusks, those of squid are rich in DAGEs (sometimes referred to as MADAGs, monoalkyldiacylglycerols). The number of molecular species of the DAGEs identified in the digestive glands of the squid *B. magister* is almost 90 [32]. Most molecular species contain a 16:0 or 18:1 alkyl group at the *sn*-1 position, whereas PUFAs are mainly esterified at the *sn*-2 position.

### 3.2. Cnidaria

Many cnidarian species contain symbiotic dinoflagellates (SD, microalgae of the family Symbiodiniaceae). It is obvious that the total lipidome of such cnidarians consists of both invertebrate animal lipids (WE, DAGE, GPL, and CAEP) and microalgae lipids (glycolipids and betaine lipids). Total TGs of symbiotic cnidarians are a mixture of TG molecules that originate from symbionts and the host. Up to 450 molecules have been identified in the TL of cnidarians (Table 1).

WE is an abundant storage lipid class in cnidarians. A comparison between WE profiles of three coral groups (symbiotic reef-building corals, symbiotic alcyonarians, and asymbiotic gorgonians) has shown cetylpalmitate (16:0/16:0) to be a major component in all the corals [35]. Other saturated WEs contain 30, 34, and 36 carbon atoms. More than 80% of saturated molecular species are found in the WEs of the reef-building coral *A. cerealis* [36]. In general, the unsaturated WE content (16:0/16:1, 16:0/18:1, and 16:0/20:1) of alcyonarians is higher than that of reef-building corals [35]. In contrast to symbiotic coral species, asymbiotic gorgonians contain a noticeable amount of long-chain WE molecular species (C_37_–C_41_) with an odd number of carbon atoms [35]. Dienoic WEs (16:2/16:0, 16:0/16:2, 18:0/16:2, 16:0/18:2, 18:0/18:2, 16:2/20:0, and 18:0/18:2) are among the dominant WE molecular species in the alcyonarian *Sinularia* sp. [43]. As reported, dienoic molecules are absent from the WEs of the alcyonarian *S. siaesensis* [45]. Only 17% of WEs are comprised of mono- and dienoic molecules in the zoantharian *Palythoa* sp. [50].

TG is an important storage lipid class of cnidarians. About 75% of TG molecules in the hydrocoral *M. platyphylla* contain the 22:6 acyl groups; the major molecular species are highly unsaturated TGs 22:6/22:6/22:6, 22:6/16:0/16:0, and 22:6/16:0/18:0 [49]. Lipids of the zoantharian *Palythoa* sp. contain noticeable amounts of TG molecular species with the 20:4 and 22:5 acyl groups [50]. A high level of TGs with the 18:3 and 22:6 acyl groups is also found in the reef-building coral *A. cerealis* [36]. However, the major TGs of tropical alcyonarians of the genus *Sinularia* are saturated: 16:0/16:0/16:0 and 16:0/16:0/18:0 [43,44,45]. Half the TG molecules identified in the jellyfish *R. esculentum* are composed of the 22:6 acyl group at the *sn*-1/*sn*-3 position and the 16:0 acyl group at the *sn*-2 position [52].

A considerable part of storage lipids in coral polyps is composed of DAGE [43,44,45,49,50,80]. However, the lipidome of the reef-building coral *A. cerealis* is distinguished by trace amounts of DAGE [36]. DAGE has not been reported for ascidian lipidomes [51]. The major DAGE molecular species of tropical coral polyps (hard and soft corals, hydrocorals, and zoantharians) are saturated, 16:0a/16:0/16:0 and 18:0a/16:0/16:0. The 16:0 and 18:0 alkyl groups at the *sn*-1 position, PUFA acyl group at the *sn*-2 position, and 16:0 acyl group at the *sn*-3 position dominate the unsaturated DAGE molecules. The high level of the 22:6 acyl groups is found at the *sn*-2 position of DAGE in the hydrocoral *M. platyphylla* [49] containing up to 35% of 22:6n-3 in total FA [80].

In coral polyps and jellyfish, PC is the most abundant structural GPL class, followed by PE. On the basis of MS signal intensity, the abundance of PC has been found to be significantly higher than that of other PL classes (22–35% of TL). More than 150 PL molecular species have been identified in cnidarians to date. Cnidarian GPLs are known to be rich in C_20-22_ PUFA and can contain VLCFA with 24 and 26 carbon atoms [81,82]. Unlike cold-water hydrocorals [83], tropical hydrocorals of the genus *Millepora* rapidly convert 20:4n-6 into 22:5n-6 and 20:5n-3 into 22:6n-3, which can be explained by the high activity of C_2_ elongase and Δ4 desaturase [44]. Furthermore, soft corals and jellyfish are capable of converting C_22_ PUFA into C_24_ PUFA [80,81].

PUFAs are mainly located at the *sn*-2 position of cnidarian GPL, whereas saturated FAs (SFAs) and monounsaturated FAs (MUFAs) are primarily located at the *sn*-1 position of GPLs. Except for *Millepora*, the most-studied cnidarian groups use 20:4n-6 and 20:5n-3, mainly for the synthesis of PE and PC. Due to the low levels of C_20_ PUFAs, *Millepora* hydrocorals have to use C_22_ PUFAs for the synthesis of PE and PC (22:5n-6 and 22:6n-3) and PS and PI (22:4n-6 and 22:5n-6) [47]. VLCFAs are concentrated in PS and PI of cnidarians [42,52]. For example, the major GPL molecular species in the polar lipidome of the soft coral *G. rubiformis* are PE 16:0p/20:4, PC 16:0a/20:4, PS 20:1/24:5, and PI 16:0/24:5 [41]. In the cnidarians compared, most molecular species of PE and PC are ether lipids, but diacyl molecular species dominate PI. Hydrocorals and tropical soft corals contain diacyl and ether PS molecules, respectively, whereas cold-water soft corals contain a mixture of these PS forms [47]. In general, 16:0 *N*-acyl groups dominate the CAEP molecules of cnidarians, while shorter C_14_ *N*-acyl groups dominate the CAEP molecules of the cold-water hydrocoral *A. steinegeri* [47]. The diversity of CAEP molecular species is formed by a variety of sphingoid bases but not *N*-acyl groups [30,48]. A significant part of CAEP and sterols in cnidarian structural lipids may be considered a necessary condition to form lipid rafts in cnidarian cell membranes.

Specific lipid classes of photosynthetic organisms (such as MGDG, DGDG, SQDG, and betaine lipids, e.g., diacylglyceryl-3-*O*-carboxyhydroxymethylcholine (DGCC)) are detected in the lipidomes of cnidarians containing symbiotic dinoflagellates. A complete loss of these symbionts leads to death of animals. SFAs and MUFAs such as 16:0, 18:0, and 18:1 are significantly represented in the SQDG observed [43,48,55,75]. The marker FAs of symbiotic dinoflagellates are C_16_ and C_18_ PUFAs [80]. Two C_18_ PUFAs (18:4n-3 and 18:5n-3) are well known to be concentrated in MGDGs and DGDGs [84]. Lipidomic studies have confirmed that most of MGDG and DGDG molecules contain 18:4 and 18:5 at both the *sn*-1 and *sn*-2 positions. Molecular species 18:4/18:4 and 18:4/18:5 dominate MGDG in *Millepora* [48]. In *Capnella* sp., the major DGDGs are 18:4/18:4, 18:4/20:5, and 16:2/22:6 [76]. A high level of DGDG molecules with C_20-22_ PUFA at the *sn*-1/*sn*-2 positions in *M. platyphylla* (40.8% of total DGDG) distinguishes this species from *M. dichotoma* (5.4%). This shift in the DGDG profile may be caused by the presence of other species of microalgae in *M. dichotoma* (e.g., *Chlorella*) [85]. DGCCs are mainly formed by 22:6; two VLCFAs, 28:7 and 28:8, are detected at the *sn*-2 position in some DGCC molecules [48].

### 3.3. Crustacea

Data on crab lipidomes are fragmentary and mostly concern the growth of juvenile specimens. It should be noted that an untargeted lipidomics (top-down approach) was applied in these studies, and lipid molecular species of crabs were identified using only computing software and search libraries [56]. As a result, some mistakes have appeared, e.g., the misidentification of the 25:1 and 29:1 acyl groups [59]. Up to 541 TG, 313 PC, 147 PE, and 153 ceramide molecules can be identified in the crab lipidome [62]. Several lipid classes and their molecular species (the range of carbon atoms; maximum number of double bonds) have been described from crabs: TG (42–56; 4) [57], DG (30–39; 5) [59], PE (34–42; 11), PC (29–42; 10), lysoPC (15–24; 6), lysoPE (18–24; 6), PI (36–40; 10), PS (34–40; 9), sphingomyelin (SM) (31–37; 2), and CL (62–80; 14) [58,61]. A detailed list of the lipid classes found in the crab *P. trituberculatus* is provided in the work by Yuan et al. [62]. A series of unusual TGs with VLCFAs in the crab *E. sinensis* (for example, 29:1/18:1/24:2, 26:0/18:1/23:1, 30:1/16:0/23:0, 27:1/18:1/22:1, and 27:1/18:1/22:1) was reported by Ding [59]. The presence of several molecular species of ether PL such as alkylacyl-PC, alkylacyl-PE, alkylacyl-PS, and alkenylacyl-PS was supposed [58,61]. The positional distribution of FAs in TG and PL molecules was the focus of several lipidomic studies on crabs [60,62]. In the hepatopancreas of *E. sinensis*, 16:0, 18:0, 18:1n-9, 18:2n-6, 18:3n-3, 20:4n-6, 20:5n-3, and 22:6n-3 are located at the *sn*-1,3 positions, while 16:0, 18:1n-9, 18:2n-6, and 18:3n-3 are located at the *sn*-2 position of TG molecules. Saturated 16:0 and 18:0 are located at *sn*-1 position, while 18:1n-9, 18:2n-6, 20:4n-6, 20:5n-3, and 22:6n-3 are mainly located at the *sn*-2 position of PE molecules. The same distribution of SFAs and PUFAs is observed in PC molecules, but with addition of 18:1n-9 at the *sn*-1 position. In the hepatopancreas of *P. trituberculatus*, 16:0 and 18:0 dominate acyl groups at the *sn*-1 position of TG molecules; 18:1n-9 and 18:2n-6 at the *sn*-2 position; and 22:6n-3 at the *sn*-3 position. The acyl groups 16:0, 18:0, and 18:1n-9 are located at the *sn*-1 position, whereas PUFAs (mainly 20:5n-3) are located at the *sn*-2 position of PE and PC molecules. Storage TG is the most abundant lipid class in hepatopancreas, whereas structural PE and PC are most abundant in the muscles of crabs. Similar to the studies reviewed above, SM, but not CAEP, was earlier detected in the TL of muscles and hepatopancreas of five commercially exploited crabs [86].

As regards the shrimp species studied, up to 200 PL molecular species can be counted [68,70]. The major lipid classes detected in shrimp tissues are TG, PC, and PE, followed by SM, CL, PI, PS, and phosphonolipids [66,67]. Up to 70% of TLs are comprised of TGs in the eggs of *M. borellii* [64], whereas PL classes dominate the TLs in the gills and muscle of this shrimp. Specifically, 37 PC, 50 PE, and 34 PS have been characterized in lipids recorded from three shrimp species [68]. Among them, 16:0/18:1, 16:0/20:5, 16:0/22:6, 18:0/20:5, and 18:0/22:6 are the dominant diacyl form; 16:0a/18:1, 18:1a/18:2, and 18:0a/22:5 may be the dominant alkylacyl form; and 16:0p/20:5, 16:0p/22:6, and 18:0p/22:6 may dominate plasmalogens [68]. The contents of ether forms of PC and PE ranged from 10 to 15% of the corresponding PL classes in the shrimp *P. kerathurus* [66]. Six species of alkylacyl-PE, four species of alkenylacyl-PE, sixteen species of alkylacyl-PC, and eighteen species of alkenylacyl-PC have been identified in the spiny lobster *P. vulgaris* [69]. Hansen et al. [70] reported about 99 WE and 233 TG compounds in the copepod *Calanus* sp., but neither WE nor TG molecular species were listed in their paper. Huang et al. [63] found several fantastic PUFAs such as 37:1, 38:6, 39:4, 39:5, 41:4, and 41:5 at the *sn*-1 position of PE and PC molecules, but these seem to be abbreviations of PL molecular species rather than PUFAs.

### 3.4. Echinodermata

Different animal tissues and extraction methods have been applied by different researchers in their studies of echinoderm lipidomes. Kostetsky et al. [71] used the classic Folch’s method [87] of extraction from fresh whole starfish, the intestinal tracts of sea urchins, and the muscular sacs of sea cucumbers collected by divers. Omran et al. [72] extracted the body walls of sea cucumbers with ethanol. Wang et al. [73] purchased dried sea cucumbers from a free market and carried out extraction according to Bligh and Dyer [88]. Therefore, these lipidomic data are quite heterogeneous.

Kostetsky et al. [71] described the composition of PE and PC molecular species in echinoderms from the Sea of Japan and showed that alkylacyl-PC and alkenylacyl-PE are the dominant forms, except for the sea urchin *S. intermedius* that contains 63% diacyl-PE, 7% alkylacyl-PE, and 30% alkenylacyl-PE. In the muscle tissues of echinoderms, the major PC molecular species are 18:1a/20:5, 16:0a/20:5, 18:0a/20:5, 18:1/20:5, 18:0/20:5, 20:1/20:5, and 16:0/20:5. The positional distribution of acyl groups (at *sn*-1/*sn*-2) in the PC molecules of starfish is characterized by the highest SFA/PUFA content, while sea cucumbers have the highest MUFA/PUFA content. The major PE molecules are 18:0p/20:4, 18:0p/20:5, 18:0a/20:5, 18:0/20:4, 18:0/20:5, 18:1/20:5, and 20:1/20:5. The molecular species of PE contain 63.4–84.3% SFA/PUFA and 11.9–33.4% MUFA/PUFA.

A total of 45 TG molecules with 46–60 carbon atoms and 0–9 double bonds have been found in ethanolic extracts from six species of Egyptian sea cucumbers [72]. At least 295 PL molecular species in 11 PL classes, including PC, PE, PS, PI, PA, PG, lysoPC, lysoPE, lysoPS, lysoPI, and phosphonoethanolamine, have been detected in dried sea cucumbers from Qingdao (China) [73]. The results confirm that these echinoderms are rich in ether PL, especially alkylacyl-PC and alkenylacyl-PE. The most dominant PUFAs of PE and PC are 20:5n-3 and 20:4n-6 [73]. The FAs of PS are partly composed of saturated or monounsaturated VLCFAs; however, the dominant PUFAs in sea cucumber PS are 20:2 and 22:2. The dominant molecular species of PI contain 20:4n-6. Odd-chain FA (C17-23, with an unsaturation degree of 0–1) are present in PS, PI, PC and PE, representing 50, 35, 20, and 20% of the lipid class, respectively, and mostly esterified at the *sn*-1 position of the glycerol backbone.

### 3.5. Porifera

The study of the sea sponge *A. queenslandica* (= *Reniera* sp.) [74] is considered “lipidomic” but deals with total FA and sterol profile and does not analyze lipid molecular species. However, the general approach to achieving the objectives of this study is consistent with the goal and principles of lipidomics, since an attempt is made to establish a link between the profile of lipid molecules and the presence of genes encoding enzymes for the synthesis of these lipids. Sponges are animals, but they host dense communities of microbial symbionts. Therefore, it is unclear which FAs can be synthesized by the animal de novo, and which require input from the microbial community. Similar to ghd FA profiles of other demosponges, that of *A. queenslandica* is dominated by SFAs and “demospongic” Δ5,9 VLCFA [71]. The genetic and FA repertoires of *A. queenslandica* are compared to identify which FA could be potentially synthesized and/or modified by the sponge. As regards SFA, no evidence has been found for the required fatty acid synthase-type enzyme in *A. queenslandica*, either the animal homolog (FASN) or the equivalent enzymes in fungi (Fas1/Fas2) or bacteria (fas). This sponge also appears to lack the enzyme necessary to synthesize the branched-chain FAs. However, *A. queenslandica* does contain some enzymes necessary for the downstream FA elongation. Additionally, Δ4-, Δ5-, Δ6-, and Δ9-like desaturases are present, while the sponge lacks Δ12 and Δ15 desaturases. It has been supposed that *A. queenslandica* cannot produce basic SFA de novo, but it should be able to modify certain FAs into the more complex Δ5,9 VLCFA [74].

## 4. Major Applications

In this chapter, the lipidomic studies are grouped on the basis of their main aims to show the current applications of lipidomics to study marine invertebrates. Some interesting conclusions derived from the lipidomic data are also mentioned. The major applications below are not ordered by their importance.

The first obvious application is the detailed description of the chemical structure and content of a wide range of lipid molecules of several lipid classes [8,23,24,26,27,31,32,33,34,35,41,42,45,46,47,49,50,51,52,53,54,55,61,66,68,69,73]. Lipidomic data may cover non-polar (storage), polar (structural), or all lipid classes. The computer software for top-down (short-gun, untargeted) lipidomic analysis produces a list of thousands of low-molecular-weight compounds containing hundreds of lipid molecules. Testing a correction of identification for each compound is a challenge. Bottom-up (targeted) lipidomic analysis involves the sequential determination of the detailed chemical structure of each component followed by the reconstruction of the lipidome by a researcher. Of course, more accurate data require more efforts. Therefore, many researchers limit themselves to describing only polar lipidomes. Lipidome description extends our knowledge about the biochemical diversity of marine lipids [8,23,24,26,32,34,42,45,50,73], species- and tissue-specific features [27,47,51,52,61,68], and lipid biosynthesis [41]. Some studies are conducted using both traditional lipid techniques (FA and lipid class analyses) and lipidomic techniques, which allow the accurate determination of the acyl group structures or lipid quantification [20,24,42]. The distribution of FAs between lipid classes or between the *sn*-1/2/3 positions of the glycerol backbone can show the biosynthetic relationships between lipid classes, e.g., a possible head group exchange between GPL classes [29,47]. In contrast to PE and PC, the concentration of VLCFAs in PS and PI has been measured [41]. The modification of the host lipidome by symbionts has been revealed in symbiotic cnidarians [54,55]. The large proportion of alkenylacyl-GPL in sea slug species suggests that these compounds may play a key role in the chloroplast incorporation in sea slug cells [31]. The presence of a considerable amount of long-chain WE molecular species (C37–C41) with an odd number of carbon atoms indicates that asymbiotic soft corals may use bacterial FA in the WE biosynthesis. This observation confirms the assumption that bacterial community is important for maintaining the energy balance in azooxanthellate corals [35]. The lipidomic data are associated with the anti-inflammatory, antioxidant, and cardiovascular effects of GPLs [33].

Second, lipidomic data have been applied to the chemotaxonomy of marine invertebrates, similar to the data on the classic chemotaxonomic markers such as FAs and sterols. The alkenylacyl-PS and alkenylacyl-PE profiles allow the discrimination of shellfish genera, since these GPL classes include a large number of specific FAs [12]. A chemotaxonomic study using lipidomic data has been conducted for the most abundant Egyptian sea cucumbers [72]. The study by Henry et al. [37] provides specific information on the understudied metabolome of the reef-building coral *A. cervicornis* and further confirms that differences in metabolome structure can drive phenotypic variation among genotypes.

The optimization and development of analytical methods can be considered the third application of lipidomics in the study of marine invertebrates. The presence of a high level of alkenylacyl-PS and alkenylacyl-PE with omega-3 PUFA and multiple isobaric molecular species isomers is a noteworthy characteristic of marine oysters. Simple and robust methodology should be particularly valuable for detailed characterization of marine invertebrates with respect to omega-3 aminophospholipids [9]. A sensitive and precise method for analyzing plasmalogens in shellfish and mussels that has been demonstrated [11] can be a good choice for the large-scale preparation of plasmalogens. Suitable methodologies have been developed to quickly fingerprint lipids in the Mediterranean mussel (*M. galloprovincialis*) using multidimensional and hyphenated techniques [13]. Reversed-phase ultra-high-performance liquid chromatography−mass spectrometry, which is a method typically applied in the lipidomic analysis of most biological samples, has been optimized for comprehensive lipidomic analysis of marine shellfish [17]. The profiles of GPL molecular species extracted from mussels using the Folch’s, Bligh–Dyer’s, and methyl-tert-butyl ether (MTBE) methods are compared by high-performance liquid chromatography-electrospray ionization-tandem mass spectrometry [18]. Plasmalogens of the clam *C. fluminea* have been extracted with titanium dioxide (TiO_2_) on fibrous silica nanosphere and phospholipase A1 hydrolysis, and the molecular composition of plasmalogens has been analyzed [22]. Graphene/fibrous silica is applied for solid phase extraction of clam GPLs [21].

Seasonal variations in the lipidome have been investigated in marine invertebrates. This application opens up a wide range of opportunities for the development of strategies aimed to prevent warming-related decreases in aquaculture productivity, and also for choosing a better season for harvesting to reach a higher nutritional value. Seasonal variations have been described for the lipid molecules profiles of oysters [10], mussels [15], and clams [25]. Using the example of a symbiotic tropical hydrocoral *Millepora*, it has been shown that the remodeling of alkyl and sphingosine groups in three structural lipid classes, rather than unsaturation degree of their FA groups, may be considered as a seasonal adaptive response of host biomembranes in *Millepora* [48]. Symbionts control their lipid class composition throughout the year, but their lipid molecular species composition varies in a random manner [48]. Fingerprinting of a non-polar lipidome in the scallop digestive gland shows a prevalence of low unsaturated TGd in spring, an increase in the unsaturation degree of TGs in summer, and a dominance of long-chain and highly unsaturated TGs in autumn [28].

The following application is based on studies regarding the effect of diet on the total lipidome of marine invertebrates and on the tracing of FA trophic markers (FATM) in marine food webs by lipidomic methods. Dietary lipids and FA considerably modify the lipid class and molecular species profiles in crabs [60,62]. In blue mussel spat, the dietary essential PUFAs influence not only storage lipids, but also structural lipids [20]. To study the transfer of lipids in polar marine ecosystems, FATM of soft corals was traced in the polar lipidome of a nudibranch mollusk feeding on corals [29]. It has been found that this nudibranch decomposes coral lipids and reallocates the coral FATM to PS, PE, and PC. This reallocation is supposed to be responsible for the considerable accumulation of coral FAs in nudibranch tissues. In contrast to the FATM method, the lipidomic approach explains the rearrangement of dietary FAs in predators’ lipids and the differences in biosynthetic relationships of GPL classes between the nudibranch and corals [29]. In different cold-water nudibranch species, dietary FAs are directed to PE/PC/PS and, in some cases, form isobaric GPL molecular species. In these nudibranch’s isobaric species, the length of alkyl groups is reduced when VLCFAs are obtained with diet. This molecular mechanism may explain the adaptation of nudibranch membrane structure to the dietary input of unusual VLCFAs [30].

The effect of thermal stress on the lipidome of marine invertebrates has been considered in several studies. The shotgun lipidomics strategy has shown that the hydrolysis mechanism of GPLs during seafood storage correlates with the lipid hydrolytic enzyme activities under low storage temperatures [7]. The lipidomic approach has pointed out the important role of ceramide-based lipids in the adaptation of mussels to thermal stress [14]. The profiles of lyso-forms of PE and PC do not change under conditions usually employed for seafood transportation and storage but significantly increase during more severe thermal stresses because of the death of the whole or a part of the mollusks [16]. Coral bleaching under increasing temperatures, the major cause of coral reef mortality, was studied by lipidomic methods on soft and hard corals exposed to a short-term experimental thermal stress [36,43,44]. The heat exposure immediately disturbed the daily cycle of the composition of glycolipid molecular species in symbiotic dinoflagellates of corals. It is probable that PE molecular species are a primary target of thermal stress in the coral host [36,43,44]. Kostetsky et al. [71] analyzed the composition of molecular species of PE and PC and described the thermotropic behavior of pure PE and PC of several marine invertebrates.

An important study has described the influence of contamination and environmental factors on marine invertebrate lipidomes. The changes in lipidome profiling of ZnO nanoparticles in corals exposed to Irgarol 1051 and in Cu-exposed oysters have been documented [6,39,40]. Marine copepods sampled close to offshore oil platforms display location-specific polyaromatic hydrocarbons and lipid profiles [70]. PC molecular species in the shrimp *M. borellii* change in response to exposure to a water-soluble fraction of petroleum [64]. The lipidome of deep-sea hydrothermal vent shrimps varies according to acclimation pressure [65]. The effects of low salinity, pH, and oxygen on the growth and lipidomic responses of juvenile shrimp and crabs have been studied [56,63]. The pathogenic microsporidian *Hepatospora eriocheir* contributes to lipid metabolism disorders in the crab *E. sinensis* [59].

We suggest that lipidomic analyses are very informative in investigations of embryogenesis in marine invertebrates. To date, only two articles have reported dynamics of neutral and polar lipid molecular species during the embryonic development of crabs [57,58].

Lipidomics is sometimes considered as part of metabolomics. The metabolomic approach is also applied to study marine invertebrate lipids. The metabolomic profile including free amino acids, 5′-nucleotides, TGs, and GPLs is used in studies of season- and sex-related variations in mussels [19]. Sogin et al. [38] presented a multivariate biomarker approach to assess impacts of climate change and improve mechanistic understanding of stress response (increases in temperature and ocean acidification) in corals. The metabolomic approach seems to be useful to evaluate non-specific physiological and biochemical responses of marine invertebrates to environmental or nutrition factors. However, in this case, lipids become only an ordinary group among a vast variety of low-molecular-weight substances, and we usually do not take into account the huge and very important database on FAs and lipids of marine invertebrates accumulated over decades.

## 5. Conclusions

Over the past six years, the number of publications with lipidomics applied in the study of marine invertebrates has increased dramatically, thus indicating a surge of interest in this research area. The rapid development of lipidomic techniques allows reliable identification and quantification of all lipid molecular species in marine invertebrates. The fundamental characteristics of lipidomes in some of the taxonomic groups of marine invertebrate animals are now beginning to be understood, but the number of the species studied still remains very small. Most studies were carried out on commercially exploited species. Today, lipidomic studies are based mainly on traditional lipid techniques applied to address traditional research issues such as the description of lipid composition, chemotaxonomy of marine invertebrates, seasonal variations in FAs and lipids, FA trophic markers in marine food webs, effects of environmental factors, etc. It is advisable to carry out analyses of FAs, lipid class, and the lipidome simultaneously. Close attention should be paid to the relationship between the lipidome/FAs and lipid biosynthesis pathways. It is necessary to extend the range of applications of lipidomics and develop a new robust lipidomic methodology that will not only complement traditional methods, but also bring lipidomics to a qualitatively higher level in the traditional areas of marine invertebrate research.

## Figures and Tables

**Figure 1 marinedrugs-19-00660-f001:**
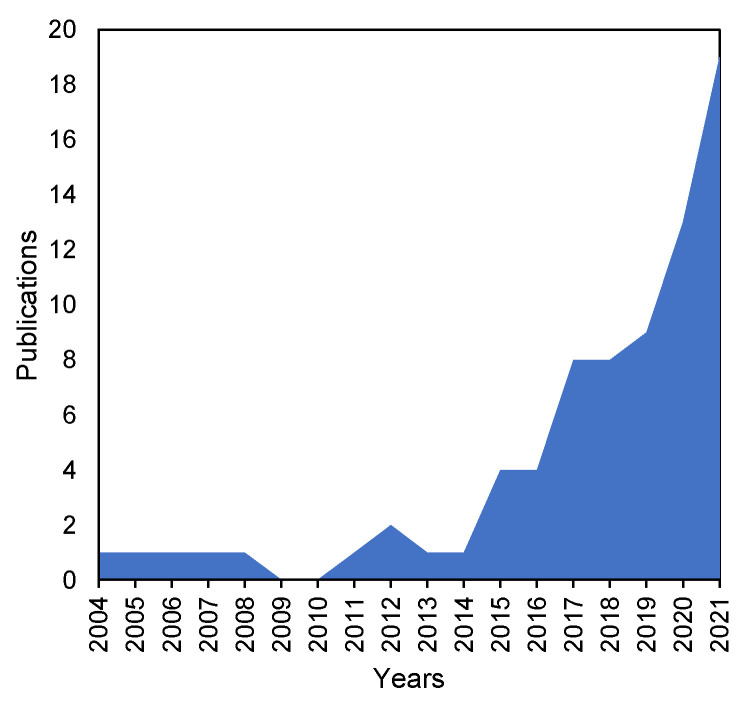
Dynamics of publications on lipidomics of marine invertebrates.

**Table 1 marinedrugs-19-00660-t001:** Marine invertebrates studied by lipidomic approach. Abbreviations: TL, total lipids; PL, polar lipids; WE, wax esters; TG, triacylglycerols; GPL, glycerophospholipids; PE, ethanolamine GPL; PC, choline GPL; PS, serine GPL; PI, inositol GPL; PlsGPL, plasmalogens; GL, glycolipids; Cer, ceramides; FA, fatty acids; FFA, free fatty acids.

Phylum or Subphylum	Animal or Tissues	Species Name	Total Molecular Species Identified	Reference
Mollusca	oyster, digestive glands	*Crassostrea hongkongensis*	1234 TL *	[6]
	oyster	*C. plicatula*	94 PL	[7]
	oyster	*C. lugubris*	90 PL	[8]
	oyster	*C. sikamea*	18 PS, 32 PE	[9]
	oyster	*C. talienwhanensis*	290 PL	[10]
	oyster	*C. gigas (=Ostrea gigas)*	59 PL, 18 PS, 32 PE	[9,11]
	oyster, edible foot	*Ostrea edulis*	62 PL	[12]
	mussel	*Mytilus galloprovincialis*	185 PC, 131 PE	[13,14,15,16]
	mussel, edible foot	*M. galloprovincialis*	29 PE, 4 PI, 5 PS, 15 PC	[12]
	mussel	*M. edulis*	230 PL, 413 TL	[11,17,18,19]
	juvenile blue mussels	*M. edulis*	343 TL	[20]
	clam	*Corbicula fluminea*	35 PL	[21,22]
	clam	*Cyclina sinensis*	435 PL	[23]
	clam	*Mactra chinensis*	453 PL	[23]
	clam	*M. veneriformis*	468 PL	[23]
	clam	*Meretrix lyrata*	96 PL	[24]
	clams	*M. meretrix*	443 PL	[23]
	clam	*Ruditapes philippinarum*	427 PL	[17,23,25]
	clam	*Saxidomus purpurata*	444 PL	[23]
	clam	*Sinonovacula constricta*	238 PL	[11,26]
	clam	*Solen gouldi*	235 PL	[26]
	clam	*Venerupis variegata*	19 PlsGPL	[11]
	clam, edible foot	*Venus gallina*	88 PL	[12]
	abalone	*Haliotis rubra*	19 PlsGPL	[11]
	abalone	*H. discus hannai*	125 TL	[17,27]
	scallop, digestive glands	*Chlamys islandica*	224 TL	[28]
	scallop	*Patinopecten yessoensis*	125 TL	[17]
	scallop	*Placopecta magellanicus*	19 PlsGPL	[11]
	nudibranch	*Tritonia tetraquetra*	90 PL	[29,30]
	nudibranch	*Dendronotus* sp.	90 PL	[30]
	nudibranch	*D. robustus*	90 PL	[30]
	sacoglossan sea slug	*Elysia viridis*	322 PL, 47 GL	[31]
	sacoglossan sea slug	*Placida dendritica*	318 PL, 14 GL	[31]
	squid, digestive glands	*Berryteuthis magister*	90 DAGE	[32]
	squid, viscera and gonads	*-*	1223 PL	[33,34]
Cnidaria	reef-building coral	*Acropora acuminate*	83 WE	[35]
	reef-building coral	*A. cerealis*	79 TL	[36]
	reef-building coral	*A. cervicornis*	922 non-polar metabolites	[37]
	reef-building coral	*A. acuminata*	83 WE	[35]
	reef-building coral	*A. cytherea*	83 WE	[35]
	reef-building coral	*Pocillopora damicornis*	450 TL	[38]
	reef-building coral	*Seriatopora caliendrum*	22 PC	[39,40]
	reef-building coral	*Turbinaria peltata*	83 WE	[35]
	soft coral	*Mopsella spinosa*	83 WE	[35]
	soft coral	*Menella flora*	83 WE	[35]
	soft coral	*M. praelonga*	83 WE	[35]
	soft coral	*Gersemia rubiformis*	68 GPL	[41]
	soft coral	*Capnella* sp.	32 GPL	[42]
	soft coral	*Cladiella laciniosa*	83 WE	[35]
	soft coral	*Sinularia* sp.	170 TL	[43]
	soft coral	*S. brassica*	83 WE	[35]
	soft coral	*S. heterospiculata*	170 TL	[44]
	soft corals	*S. macropodia*	32 GPL	[42]
	soft coral	*S. robusta*	83 WE	[35]
	soft coral	*S. siaesensis*	144 TL	[45]
	soft coral	*Xenia* sp.	32 PL	[46]
	hydrocoral	*Millepora dichotoma*	152 PL	[47,48]
	hydrocoral	*M. platyphylla*	179 TL	[47,48,49]
	hydrocoral	*Allopora steinegeri*	127 PL	[47]
	zoantharian	*Palythoa sp.*	145 TL	[50]
	ascidian	*Ciona intestinalis*	245 TL	[51]
	ascidian	*Halocynthia roretzi*	245 TL	[51]
	ascidian	*Styela clava*	245 TL	[51]
	jellyfish	*Rhopilema esculentum*	75 PL, 12 TG	[52]
	jellyfish	*Phyllorhiza punctata*	12 PL	[53]
	sea anemone	*Aiptasia pallida*	109 PL	[54,55]
Crustacea	crab	*Cancer magister*	195 TL	[56]
	crab embryos	*Carcinus maenas*	98 PL, 16 TG	[57,58]
	crab hepatopancreas	*Eriocheir sinensis*	67 TL	[59,60]
	crab embryos	*Necora puber*	98 PL, 16 TG	[57,58]
	crab, viscera and muscles	*Portunus trituberculatus*	250 PL	[61]
	crab, viscera and muscles	*Portunus pelagicus*	250 PL	[61]
	crab hepatopancreas	*P. trituberculatus*	541 TG, 313 PC, 153 Cer, 147 PE	[62]
	juvenile shrimp	*Litopenaeus vannamei*	196 TL	[63]
	shrimp	*Macrobrachium borellii*	15 PC	[64]
	shrimp head	*-*		[34]
	shrimp	*Mirocaris fortunata*	44 PE, 44 PC, 15 SM	[65]
	shrimp, muscle and cephalothorax	*Penaeus kerathurus*	FA of PL	[66,67]
	shrimp, head and body carapace	*P. monodon, P. vannamei, P. chinensis*	200 PL	[68]
	lobster, muscle and cephalothorax	*Palinurus vulgaris*	40 PE and PC	[69]
	copepods	*Calanus* sp.	99 WE, 233 TG *	[70]
Echinodermata	sea cucumber	*Apostichopus japonicus*	29 PE, 26 PC	[71]
	sea cucumber	*Actinopyga mauritiana*	45 TG, 15 FFA	[72]
	sea cucumber	*Bohadschia marmorata*	45 TG, 295 PL	[72,73]
	sea cucumber	*Cucumaria frondosa japonica*	295 PL	[71,73]
	sea cucumber	*Eupentacta frau datrix*	29 PE, 26 PC	[71]
	sea cucumber	*Isostichopus fuscus*	295 PL	[73]
	sea cucumber	*Holothuria atra*	45 TG, 15 FFA	[72]
	sea cucumber	*H. edulis*	45 TG, 15 FFA	[72]
	sea cucumber	*H. leucospilota*	45 TG, 15 FFA	[72]
	sea cucumber	*H. mexicana*	295 PL	[73]
	sea cucumber	*H. polii*	45 TG	[72,73]
	sea cucumber	*Parastichopus califormicus*	295 PL	[73]
	starfish	*Distolasterias nipon*	29 PE, 26 PC	[71]
	starfish	*Asterias amurensis*	29 PE, 26 PC	[71]
	sea urchin	*Strongylocentrotus intermedius*	29 PE, 26 PC	[71]
Porifera	sea sponge	*Amphimedon queenslandica*	37 FA	[74]

* A total list of lipid molecular species is not available.

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
