# Peer review of "Current Progress in Lipidomics of Marine Invertebrates"

_marinedrugs, 2021, doi:10.3390/md19120660_

Round 1
Reviewer 1 Report
This is an interesting work that gathers up-to-date information on the lipidomics of marine invertebrates. Its greatest virtue is that it provides an update of relevant information on the subject. A brief introduction is followed by an account of the phyla studied and the lipidomics performed. A third part concentrates on the lipidomic data, whereas the fourth deals with what the author calls “Major applications” and further refers to as “trends”. I would suggest a reconsideration of the term trend and just use application. The paper ends with a conclusions section. In my opinion, the greatest weakness of the work lies in the language. The manuscript needs a substantial review by an English speaking specialist. Many of the sentences and or paragraphs combine verbs in present and past tense, and some are even difficult to understand (for example L90, 105, 116, 163…..). I also find an inconsistency with the definition and use of the term lipidomics. Defined at the introduction as “the complete lipid profile of a biological object” it is latter used to describe the partial lipid composition of some organisms (for example L 402 or L458) or redefined according with the “goal and spirit” of the term (L 425). Besides, although the concepts of target and shotgun approaches are dealt with along some parts of the text, a consideration and definition of such approaches in the introduction would improve the paper. Another improvement would be the use of a list of abbreviations and to unify the terms of such abbreviations. For example, PC and PE are further referred to as PtdCho and PtdEtn (L172). I would suggest also to reconsider the term “Biological objects” since object describes a material thing. Finally, the sentence in L 576 does not seem to me to be a valid conclusion since it has not been dealt with along the manuscript (the direct comparison marine/terrestrial).
Author Response
The manuscript has been revised to improve the language.
Undoubtedly, lipidome is “the complete lipid profile”, but lipidomicsis large-scale study of composition, metabolism or functional role of lipids. Works with a partial description of the molecular species profile fit the definition of targeted lipidomics. As for the article in the "Porifrera" section, there are currently no works devoted to the large-scale study of the molecular species of lipids in sponges. However, it would be wrong not to consider this group of invertebrates.
A brief description of the main lipidomic approaches has been added to the introduction.
The journal rules do not provide for a list of abbreviations in publications. Abbreviations for lipid names have been unified.
Term “Biological objects” changed to “research objects” or “organisms”.
We have removed comparison of marine/terrestrial organisms from conclusions
Reviewer 2 Report
Reviewer’s comments
The manuscript by Dr. Imbs is of great interest because it provides a complete survey of the scientific literature on lipidomic studies applied to marine invertebrates during the recent years since 2004.
Thus, this review will be a valuable tool for researchers starting lipidomic studies or already working in this field because it provides comprehensive information and very usefully classified.
The objectives of this review are particularly laudable. The manuscript is clearly written. I have a positive feeling for the paper in the present state.
In conclusion, I consider that the paper could be published in the present state after some slight improvements.
- In the Abstract and the Introduction, I believe that two points do not seem explicit enough and needs to be clarified even at this stage.
a) Really, why is it important to conduct research with a lipidomic approach? What are looking for in searching lipidomic features of an invertebrate class? What kind of results are expected comparatively to those given by the “classical lipidology“?
What kind of potential applications are searched or expected ? “Classical lipidology“ has indeed produced a large number of results in marine lipids as reported briefly in the Conclusions. Is there a positive link between the two approaches? This question seems not enough clear in the text.
b) The reader will ask why the chose 2004 as the starting point of the review. Is there a specific reason?In line with figure 1, the production of publications on lipidomics of marine invertebrates remains constant from 2004 to 2008. But then how was it in just before these years?
- I would also suggest introducing the most important aspects developed in the “Major Applications” chapter towards the end of the introduction.
- The expression « Due to a rapide development of …“ seems indeed not enough (Lines 39-43). The author could point out in a few lines if major advances in analytical techniques and instrumentation occurred in the period 2004-2021.If so, which ones, please.
4. Lines 558-561.
If “only two articles have reported dynamics of neutral and polar lipids molecular species…“ it seems difficult for the Author to speculate here on the interest of lipidomic analyses in the studies of embryogenesis.
- In the text, the author considers an important point concerning the validity of lipidomic analyses in the case of marine invertebrates hosting an important bacterial biomass such as sponges.Maybe, this aspect deserves a little more development.
- It would be of interest to provide an abbreviation list according to the procedures of the Journal.
Author Response
We have included information on the benefits and application of the lipidomic approach in the introduction. In our opinion, there is no need for a more detailed comparison of classical lipidology and lipidomic approach in current review.
Han and Gross first defined the concept of lipidomics in a review article in 2003 [3], so 2004 was chosen as the starting point.
The main aspects of lipidomics application and description of new mass-spectrometry equipment and methods have been added to the introduction section.
One of the purposes of this review is to highlight all the applications of lipidomics. Therefore, in our opinion, it would be wrong not to mention the possibility of using the lipidomic approach in the study of embryo and ontogenesis.
All invertebrates contain a certain amount of symbiotic or parasitic microorganisms, which makes it difficult to analyze the host lipidome, and sponges are the undisputed leaders here. However, we were unable to find studies on the microbial communities of sponges using the lipidomic approach (there are several studies using fatty acid markers).
The journal rules do not provide for a list of abbreviations in publications.
Round 2
Reviewer 1 Report
This new revised version of the paper reads much better, however I still have two minor concerns. There are minor English spell checks required (for example in L6 (has), L51 (was), 123 (studied), 144-145 (have mainly dealt with)). In L 603 there is still a direct reference to the lipidome of terrestrial animals. As mentionned in the first review this has not been dealt with in the paper, and in my opinion should removed. The authors mention in their response that they have done so, but the sentence is still there.
I would like to send my condolences to the team for the loss of Prof. Imbs. A remarkable scientist.
Author Response
The manuscript has been re-revised to improve the language.
We have finally removed the marine / terrestrial comparison from the conclusions.